# MODEL MERGING WITH SVD TO TIE THE KNOTS

**George Stoica**[1*]    **Pratik Ramesh**[1*]    **Boglarka Ecsedi**[1]
**Leshem Choshen**[2]    **Judy Hoffman**[1]

[1]Georgia Tech    [2]IBM Research, MIT

Correspondence emails: {gstoica3,pramesh39}@gatech.edu

## ABSTRACT

Recent model merging methods demonstrate that the parameters of fully-finetuned models specializing in distinct tasks can be combined into one model capable of solving all tasks without retraining. Yet, this success does not transfer well when merging LoRA finetuned models. We study this phenomenon and observe that the weights of LoRA finetuned models showcase a lower degree of alignment compared to their fully-finetuned counterparts. We hypothesize that improving this alignment is key to obtaining better LoRA model merges, and propose KnOTS to address this problem. KnOTS uses the SVD to jointly transform the weights of different LoRA models into an aligned space, where existing merging methods can be applied. In addition, we introduce a new benchmark that explicitly evaluates whether merged models are general models. Notably, KnOTS consistently improves LoRA merging by up to 4.3% across several vision and language benchmarks, including our new setting. We release our code at: https://github.com/gstoica27/KnOTS.

## 1 INTRODUCTION

Model merging (Garipov et al., 2018; Draxler et al., 2018; Wortsman et al., 2022a; Choshen et al., 2022) is an increasingly popular technique that can surprisingly create a single general model by combining weights of task-specific models. This allows creating multi-task models by accumulating skills (Stoica et al., 2024; Ilharco et al., 2023; Yadav et al., 2023; Ortiz-Jimenez et al., 2024), a desirable trait in various scenarios such as recycling models shared on hubs (Choshen et al., 2023), patching model weaknesses (Cai et al., 2023; Zaman et al., 2023) and collaborating to improve models (Don-Yehiya et al., 2023).

Model merging approaches have found substantial success when merging models that are full-rank finetuned (i.e., all parameters are tuned with maximum rank, we denote it as FFT) to solve distinct tasks from the same pretrained checkpoint, into one model capable of solving all (Ilharco et al., 2023; Matena & Raffel, 2022; Yadav et al., 2023; Ortiz-Jimenez et al., 2024; Wortsman et al., 2022b). In many cases, performing a linear sum over the finetuned model weights without further training, can achieve strong performance(Wortsman et al., 2022b; Ilharco et al., 2023; Yadav et al., 2023).

Interestingly, existing merging approaches do not always transfer well when applied to models finetuned with parameter efficient finetuning (PEFT) strategies (Tang et al., 2024), such as the extremely popular Low-Rank Adaptation (LoRA) (Hu et al., 2022). We probe into this phenomenon (§ 3) by comparing the pairwise centered kernel alignment (CKA) (Kornblith et al., 2019) between FFT models and their equivalent LoRA finetuned counterparts. While we observe that FFT models have very high CKA—indicating that the finetuning-updates they apply to the respective pretrained weights (we call these "task-updates") have high alignment, LoRA models exhibit considerably lower CKA. This suggests that the task-updates between different LoRA models process inputs through misaligned subspaces. We hypothesize that aligning these task-updates is crucial to strengthening model merging operations, and propose a *data* and *gradient free* approach to do this.

Our method, termed KnOTS (*Kn*owledge *O*rientation *T*hrough *S*VD), builds upon singular value decomposition (SVD) to transform the task-updates of different LoRA models into a shared space, where merging methods can be applied (Fig. 1). Notably, KnOTS is simple and readily slots into

---

*Equal Contribution & Corresponding Authors.

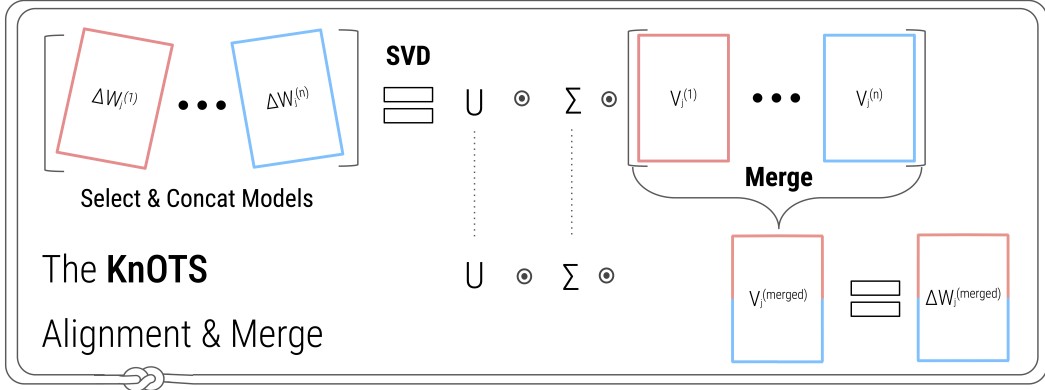

Figure 1: **The KnOTS method** for merging "task-updates" from an arbitrary layer-$j$ of different models. Each weight-update is denoted by $\Delta W_j^{(i)}$, where $i$ is the task-update of the $i^{th}$ model. KnOTS first concatenates the updates together and applies the SVD, to obtain $U, \Sigma$ and a set of concatenated $V^{(i)}$ matrices that each correspond to a particular task. KnOTS then merges the $V$'s into a single $V^{(merged)}$ matrix. Finally, KnOTS multiplies the $U, \Sigma$ and $V^{(merged)}$ to obtain a merged-update to be added to the pretrained model.

many existing merging methods. KnOTS first concatenates all the task-updates for a layer, and then decomposes the result with the SVD to obtain: $U\Sigma V^T$ (see §4). We observe that in this construction, $U$ consists of vectors that form the orthonormal basis for a *shared* representation space between all task updates, the non-negative diagonal values in $\Sigma$ represent the scale associated with each basis vector in the aligned space, and $V^T$ is a set of concatenated matrices (one for each task-update) that are all aligned to the common $U$. KnOTS then applies existing merging methods, such as those of Ilharco et al. (2023); Yadav et al. (2023), to the $V^T$ task-matrices and obtains a $V^{(merged)}$ matrix that is still aligned to the common $U$. After merging, KnOTS constructs the merged model by multiplying the resulting $U\Sigma V^{(merged)}$ into one matrix at every layer, and adding it to the corresponding parameter matrix from the pretrained model. We validate KnOTS on several model merging settings spanning vision and natural language, demonstrating that KnOTS strengthens existing model merging approaches by up to 4.3% (§ 5).

In addition, we introduce a new variant of the popular eight-vision task merging benchmark originally proposed by Ilharco et al. (2023), aimed at explicitly evaluating a merging method's ability to create *general* models. Specifically, we transform the benchmark to a *"joint-evaluation"* setting. This consists of evaluating the performance of arbitrary merged models on the *union* of all inputs and labels across all eight datasets, without providing the model with any information regarding the dataset a particular example comes from. It contrasts to the existing setting where only labels pertaining to the dataset an example stems from are given to the models to classify. We argue that the joint-evaluation setting is a step towards simulating the ability to create general merged models, capable of preserving the union of all skills captured by their base models (§ 5.3). Notably, KnOTS further improves merging performance in this challenging setting.

## 2 RELATED WORK

**Landscapes between models.** The geometric landscape of non-convex loss functions used to train neural networks remains largely uncharted. However, Draxler et al. (2018); Garipov et al. (2018); Simsek et al. (2021); Frankle & Carbin (2019) reveal that the parameter values of independently trained neural networks can be interpolated without increasing the test loss, a phenomenon known as mode-connectivity. Neyshabur et al. (2020), shows that models are often mode-connected when they are finetuned on the same task from the same pretrained initialization. In parallel, it has become common practice to finetune models using pretrained weights (Dosovitskiy et al., 2021; Huang et al., 2023; Oquab et al., 2024; Hsu et al., 2021; Touvron et al., 2023; Radford et al., 2021). Together with recent discoveries Wortsman et al. (2022b); Matena & Raffel (2022) that even the parameters of models finetuned on different tasks from the same initialization may be merged to create strong multitask models, there has been a surge of model merging approaches for finetuned models.

**Model merging methods.** Several approaches improve robustness or scores by averaging finetuned weights (Choshen et al., 2022; Wortsman et al., 2022b; Rame et al., 2022; 2024). Task arithmetic (TA) (Ilharco et al., 2023) first subtracts the parameter values of the pretrained model from those of the finetuned models, creating a set of "task-vectors." These are then linearly summed to create a merged model, without any gradient calculations or retraining. Ortiz-Jimenez et al. (2024) theoretically grounds TA, by showing that its success is dependent on the task-vectors governing disjoint regions in the pretrained model's function space—a concept known as "weight disentanglement." In practice, they find that the weights of different finetuned models heavily conflict, and propose a finetuning strategy that disentangles finetuned models to merge better with TA. Daheim et al. (2024); Shah et al. (2024) also find that TA-style merging can improve with finetuning and access to large amounts of training data. TIES (Yadav et al., 2023) improves TA by resolving the parameter interference between models when merging. Specifically, it prunes low-magnitude weights and then only averages weights which share the dominant sign. DARE (Yu et al., 2024) further explores this issue by randomly dropping fine-tuned weights and rescaling the remaining ones to create sparse task-vectors.

**Merging LoRA models.** LoRA (Hu et al., 2022) has quickly become a very popular finetuning technique. However, existing merging methods (Ilharco et al., 2023; Yadav et al., 2023; Yu et al., 2024) do not transfer well to merging LoRA models (Tang et al., 2024). Tang et al. (2024) suggest this is due to increased weight-entanglement between the models, and propose to a finetuning method similar to Ortiz-Jimenez et al. (2024) which improve the performance of each merging method. To the best of our knowledge, we present the first framework capable of achieving strong merged LoRA models *without finetuning*.

**SVD-based LoRA Approaches.** Employing the SVD on LoRA models is not a new idea for purposes other than merging and aligning weights. Meng et al. (2024) initialize the parameters of LoRA models using the SVD of the weights of the corresponding pretrained model, yielding improved finetuning performance. Zhang et al. (2023) use the SVD to improve the gradient updates in parameter efficient finetuning. The popular Hugging Face library (Mangrulkar et al., 2022) includes several merging methods for LoRA models that are suffixed by "-svd" (e.g., "TIES-svd"). These first merge the LoRA parameters of different models according solely to the prefix merging-method (e.g., "TIES"). Afterwards, they use the SVD to decompose the merged parameters back into LoRA weights. KnOTS is unrelated to these approaches. Instead, it significantly improves merging performance by aligning LoRA models with the SVD.

## 3 BACKGROUND AND MOTIVATION

**Problem setting.** We study *gradient-free* multitask model-merging (Ilharco et al., 2023; Stoica et al., 2024; Yadav et al., 2023; Jin et al., 2023). Suppose that we have a set of $n$ models with the same architecture, that have each been finetuned with Low-Rank Adaptation (LoRA) (Hu et al., 2022) on a distinct task (e.g., different image classification problems) from the same pretrained model. Let $\{f^{(1)}, f^{(2)}, \ldots, f^{(n)}\}$ denote the finetuned models and $f^{(pt)}$ be the pretrained model. Model-merging methods fuse the parameters of these finetuned models into a single unified model capable of solving all tasks. We study gradient-free approaches as they are both *data-efficient* and *training-free*, making them lightweight tools for merging models on-the-fly (Wortsman et al., 2022a). We study models finetuned with LoRA due to its wide usage and because recent work observes notable challenges when applying existing gradient-free methods on such models (Tang et al., 2024).

**Model definitions.** Let a pre-trained model $f^{(pt)}$ have $l$ layers, each containing weights and an optional bias. Assume all biases are concatenated in respective weights, and let weight of the $j^{th}$ layer be represented as $W_j^{(pt)}$. Let $\theta^{(pt)}$ be the collection of all model weights, such that $\theta^{(pt)} = \{W_1^{(pt)}, \ldots, W_j^{(pt)}, \ldots, W_l^{(pt)}\}$. Finetuned models are obtained by applying task-updates to each weight in $\theta^{(pt)}$. We denote these updates by $\tau^{(i)} = \{\Delta W_1^{(i)}, \ldots, \Delta W_j^{(i)}, \ldots, \Delta W_l^{(i)}\}$, where $\Delta W_j^{(i)}$ is the update corresponding to the $j^{th}$ layer in the $i^{th}$ model. As LoRA constrains each update to be low-rank, we let all updates have a rank of $r << \min(I, O)$, where $I$ and $O$ represent the input and output dimension. Every $\Delta W_j^{(i)} \in \mathbb{R}^{O \times I}$ transforms input features $x^{(i)} \in \mathbb{R}^I$ to output features $y^{(i)} \in \mathbb{R}^O$ as follows: $y^{(i)} = \Delta W_j^{(i)} x^{(i)}$. Due to the low-rank constraint, $\Delta W_j^{(i)}$ can only act on

a subspace of size $r << \min{(I,O)}$ over $y^{(i)}$ (Hu et al., 2022). Finally, the parameters of $f^{(i)}$ are given by $\theta^{(pt)} + \tau^{(i)} = \{W_1^{(pt)} + \Delta W_1^{(i)}, \ldots, W_j^{(pt)} + \Delta W_j^{(i)}, \ldots, W_l^{(pt)} + \Delta W_l^{(i)}\}$.

**Merging models.** Existing methods merge models by first defining a function (e.g., identity in (Ilharco et al., 2023)) that transforms the task-updates $\{\tau^{(1)}, \ldots \tau^{(n)}\}$. Second, each update is uniformly scaled via a unique "scaling coefficient", denoted by $\alpha^{(i)} \geq 0$. Third, these updates are summed to create a single set of merged updates, which are then added to the pretrained model weights to obtain the final merged model. Optimal values for $\alpha(i) \geq 0$ are selected according to the performance of the merged model on any available data Ilharco et al. (2023); Yadav et al. (2023); Yu et al. (2024). Although LoRA updates factorize into smaller matrices $A$ and $B$, we utilize the full-matrix representation $\Delta W_j^{(i)} = BA$ in all merge operations. We explain this decision in App. A.

### 3.1 LoRA MODELS ARE DIFFICULT TO MERGE

Existing full-rank finetuned (FFT) model merging approaches perform well without additional data, even on different tasks (Ilharco et al., 2023; Yadav et al., 2023). However, Tang et al. (2024) observe that the same merging methods encounter significant challenges when merging LoRA finetuned models, even when these are finetuned on the *same* tasks. Previous works (Ilharco et al., 2023). Ilharco et al. (2023); Yadav et al. (2023); Tang et al. (2024) use "task-vector orthogonality" as a proxy for understanding when merging is difficult. This consists of flattening the updates $\{\tau^{(1)}, \ldots \tau^{(n)}\}$ into a vector, and computing the pairwise cosine similarity between them. The intuition is that when the cosine similarity between vectors is near zero (i.e., they are orthogonal), their weights are disentangled and lie in distinct subspaces—enabling them to be merged without interference.

**Task-vector orthogonality may not reliably measure merge potential.** To see why, consider a simple toy example. Let $f^{(1)}(x) = W_2^{(1)}\text{ReLU}(W_1^{(1)}x)$ and $f^{(2)} = W_2^{(2)}\text{ReLU}(W_1^{(2)}x)$ be two classification models that transform inputs $x \in \mathbb{R}$ to some $y^{(1)}, y^{(2)} \in \mathbb{R}$ respectively, where $W_1^{(1)}, W_2^{(1)}, W_1^{(2)}, W_2^{(2)} \in \mathbb{R}$. Let $W_2^{(1)} = 1, W_1^{(1)} = 1, W_2^{(2)} = 1$, and $W_1^{(2)} = -1$, and assume that each weight was initialized with 0. Here, $f^{(1)}$ classifies all $x \in \mathbb{R}^+$ as positive (i.e., $y^{(1)} > 0$) and all $x \in \mathbb{R}^-$ as negative (i.e., $y^{(2)} \leq 0$), while $f^{(2)}$ does the opposite. Calculating the task-vectors of $f^{(1)}$ and $f^{(2)}$ give $[1,1]$ and $[-1,1]$ respectively—making them *orthogonal*. However, it is trivial to see that merging their model weights catastrophically affects the performance of *at least one* model. Any scaled sum over their weights with an $\alpha^{(i)}$ will preserve one model's predictions while *flipping* the other's. Despite having orthogonal task-vectors, merging the weights of these models is catastrophic. Thus, additional measures may be helpful to understand when models can be merged.

**Merging is easier when model layers share similar activations.** Entezari et al. (2022); Ainsworth et al. (2023); Jordan et al. (2023); Stoica et al. (2024) extensively show that models whose layers extract similar intermediate activations for the same inputs, are easier to merge—even when trained on distinct tasks (Stoica et al., 2024). This often occurs when the weights across models extract the same kinds of meanings in a *similar* order from the inputs (Entezari et al., 2022; Stoica et al., 2024), and are thus *aligned*. We posit that understanding model alignment from this perspective can serve as a helpful tool for understanding the challenges behind merging. One way to measure the possibility of this is using the centered kernel alignment (CKA) measure Kornblith et al. (2019). CKA measures the similarity between the intermediate activations of two models, at each layer. CKA values close to one indicate that model parameters can be transformed to process information similarly—and thus transform inputs in a *functionally aligned* manner. When CKA is low, the reverse is true.

**FFT models have very aligned CKA representations.** Fig. 2a shows the average pairwise CKAs between the task-updates of the FFT models presented in Ilharco et al. (2023). We observe very high CKA, indicating that the updates between models process information in a way that ensures the intermediate feature activations between their layers are well aligned. This suggests that their updates extract similar information from inputs at each layer, and that merging them is less likely to cause significant disruption—as seen by (Ilharco et al., 2023).

**LoRA models have considerably less aligned CKA representations.** Conducting the same CKA analysis on LoRA variants of the same models from (Ilharco et al., 2023) shows that the models

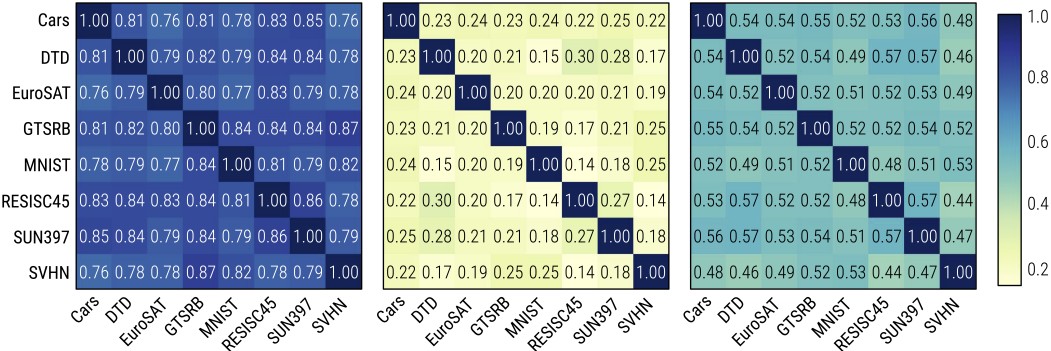

(a) Full-rank finetuned model alignments    (b) LoRA finetuned model alignments    (c) LoRA finetuned model alignments with KnOTS

Figure 2: **Finetuning strategy impacts representation alignments between models trained on different tasks**. The figure shows the average pairwise centered kernel alignment (CKA—Kornblith et al. (2019)) between the outputs *solely given by finetuning updates* (e.g., a $\Delta W_j^{(i)}$) across every attention layer, from models finetuned on different tasks with different strategies (defined in §5.2). High CKA indicates that the task-updates of different models are aligned. (a) Full-rank finetuned models exhibit high CKA and alignment.(b) LoRA finetuned models are drastically less aligned. (c) However, they are dramatically more aligned with KnOTS.

exhibit dramatically lower average CKA between task-updates (Fig. 2b). This suggests that they are functionally *misaligned*: from the same inputs each LoRA extracts unrelated features. Entezari et al. (2022); Jordan et al. (2023); Stoica et al. (2024) observe that merging models in these circumstances can yield poor performance. We speculate that this phenomenon may be due to the low-rank constraint imposed on LoRA models. Specifically, each task-update can only act on a subspace of the input and output activation space. Since each LoRA model is finetuned on a different dataset, it must carefully pick the subspace it uses to extract information. Thus, it is unlikely that the task-updates across different models extract the same kind of information, leading to low activation alignment between updates and poor merges.

**Our hypothesis.** Coupling the CKAs of FFT model task-updates with those of LoRA models, we hypothesize that aligning LoRA task-updates is a crucial first step to improving their mergeability, and the performance of existing merging methods. We seek to design such an alignment method without assuming access to data or gradients.

## 4 METHOD: KNOTS

We propose a data-free and gradient-free method for aligning LoRA models, to improve the performance of existing model merging methods. Our method, termed KnOTS (*Kn*owledge *O*rientation *T*hrough *S*VD), uses singular value decomposition (SVD) to jointly align the representation spaces between LoRA models (see illustration in Fig. 1).

**Subspace alignment with the SVD.** KnOTS aligns the updates across different LoRA models layer-wise. Thus, it suffices to consider how we align the updates for an arbitrary layer $j$. Recall that these updates are denoted by: $\Delta W_j^{(1)}, \ldots, \Delta W_j^{(i)}, \ldots, \Delta W_j^{(n)}$, each extracting different kinds of information. For notational ease, we drop the subscript $j$ for the remainder of this section. Most often SVD is used with square matrices, aligning vectors in them. Instead, we propose to align these updates into a *shared* space where they can extract *similar* information, by concatenating the updates across tasks and deconstructing the result with SVD:

$$SVD\left[\Delta W^{(1)}; \Delta W^{(2)}; \ldots; \Delta W^{(n)}\right] = U\Sigma V^T \tag{1}$$

$$= U\Sigma\left[V^{(1)}; V^{(2)}; \ldots; V^{(n)}\right]^T \tag{2}$$

where, $U \in \mathbb{R}^{O \times k}, \Sigma \in \text{Diag}(\mathbb{R}^k), \{[V^{(1)}]^T, \ldots, [V^{(n)}]^T\} \in \mathbb{R}^{k \times I}, 1 \leq k \leq \min(I, O, nr)$. Here, $\Delta W^{(1)} = U\Sigma[V^{(1)}]^T; \Delta W^{(2)} = U\Sigma[V^{(2)}]^T; \ldots; \Delta W^{(n)} = U\Sigma[V^{(n)}]^T$. With the SVD,

each task-update is decomposed into a shared $U\Sigma$ term—defining the orthogonal basis for the activations from all task-updates—along with task specific components, $[V^{(i)}]^T$. By construction, each $[V^{(i)}]^T$'s from any task-update acts to transform the *same* information (given by $U\Sigma$) with any input, thereby increasing their *alignment*. We visualize this effect in Fig. 2c. Specifically, we compute the CKA between the intermediate activations obtained from passing the same inputs through each $[V^{(1)}]^T, \ldots [V^{(n)}]^T$ of our LoRA models, and averaging over all layers to be merged. We observe that the CKAs are *significantly* higher compared to Fig. 2b, indicating that the task-updates between LoRA models are substantially more aligned under KnOTS.

**Applying merge methods.** In many cases, we can directly apply existing merging methods on the aligned $[V^{(1)}]^T, \ldots [V^{(n)}]^T$ without modifications. Specifically, we can apply some merging function (e.g., weighted sum a-la-(Ilharco et al., 2023)) over these parameters to obtain a single merged $[V^{(merged)}]^T$. We can then construct the merged update with the remaining SVD components: $\Delta W_j^{(merged)} = U\Sigma[V^{(merged)}]^T$, which is finally added to the pre-trained layer weights $W_j$.

## 5 EXPERIMENTS AND RESULTS

We validate KnOTS across diverse benchmarks spanning both vision and language domains. We first evaluate KnOTS on the popular "per-task" setting across both vision and language tasks (§ 5.2). Here, models finetuned on distinct datasets are merged and then evaluated on each dataset separately. KnOTS consistently improves the capabilities of existing merging methods across all experiments. Second, we study the capabilities of merging methods building general models by introducing a new benchmark (§ 5.3). Despite being very challenging, KnOTS is still capable of uplifting existing merging approaches. Third, we conduct extensive analysis on different facets of KnOTS (§ 5.4).

### 5.1 EXPERIMENTAL DETAILS

**Models.** We use ViT-B/32 or ViT-L/16 (Dosovitskiy et al., 2021) for all our vision experiments. These are two variants of the CLIP vision encoder (Radford et al., 2021) that are finetuned separately on a variety of tasks. Our natural language experiments are conducted with LLama3-8B model (AI, 2024) across different natural language inference (NLI) tasks. As in (Hu et al., 2022), we only apply LoRA on the weight matrices in the attention layers: namely the key, query, value and output layers. Unless otherwise specified, each LoRA has a rank of 16. Further training details are found in App. D.

**Merging methods.** As discussed in § 3, we restrict our scope to gradient-free merging methods in this work. To the best of our knowledge, there are only four such merging methods for our setting: RegMean (Jin et al., 2023), Task-Arithmetic (TA) (Ilharco et al., 2023), TIES (Yadav et al., 2023), and DARE (Yu et al., 2024). RegMean (Jin et al., 2023) is a merging approach that also *aligns* the weights of each model. It does this by solving a closed-form locally linear regression problem at every model layer. This results in obtaining a transformation matrix for each models' task-updates, that when applied aligns the updates between models. In contrast, TA, TIES and DARE assume that the task updates between models stemming from a shared pre-trained checkpoint are functionally aligned and directly merge them.

TA (Ilharco et al., 2023) merges models finetuned on different tasks with the same pretrained checkpoint by linearly summing their parameters. They apply a summation-weight (termed "scaling coefficient") to the parameters of each model that achieves the best merged model performance over a held-out validation set. Instead of tuning a unique scaling co-efficient for each task-vector which may be expensive and does not scale well, we only tune a single scaling co-efficient for all models as recommended by Ilharco et al. (2023); Yadav et al. (2023). TIES (Yadav et al., 2023) extends TA by reducing noise and conflicts between parameters when merged. It reduces noise by pruning $k$ parameters with the lowest magnitudes, and reduces parameter conflicts by also pruning those that lie in outlying directions—a step called "sign-resolution". TIES then linearly sums the model parameters using the scaling coefficients and $k$ for pruning, that achieve the best merged model over the held-out set. DARE (Yu et al., 2024) reduces noise by randomly pruning parameters following a Bernoulli distribution with probability $p$. The remaining parameters are then rescaled by $1/(1-p)$ to account for scale lost from pruned parameters. Afterwards, DARE typically employs sign-resolution of TIES (specified by DARE-TIES). To account for its random pruning step, we run

Table 1: **Eight models per-task results.** We merge eight ViT-B/32 models finetuned with LoRA on different image classification datasets. "Finetuned" refers to the accuracy of each finetuned model on the dataset it was trained on. We report the per-task (including the average) normalized-accuracies across other merging baselines. These describe how close they get to the "Finetuned" accuracy. KnOTS-TIES improves over baselines by up to 4.3% average accuracy.

| Method | Datasets | | | | | | | | |
|---|---|---|---|---|---|---|---|---|---|
| | Cars | DTD | EuroSAT | GTSRB | MNIST | RESISC45 | SUN397 | SVHN | Avg |
| | Per-Task Absolute Accuracies (%) | | | | | | | | |
| Finetuned | 74.0 | 58.3 | 99.0 | 92.7 | 99.3 | 88.4 | 64.5 | 96.2 | 84.1 |
| | Per-Task Accuracies of Merged Models Normalized Against Finetuned Models (%) | | | | | | | | |
| RegMean | 80.2 | 71.3 | 37.9 | 47.3 | 43.1 | 70.5 | 93.9 | 43.0 | 60.9 |
| TA | 82.0 | 73.6 | 48.8 | 42.1 | 53.1 | **71.5** | 97.5 | 41.2 | 63.7 |
| TIES | 82.2 | 72.8 | 50.0 | 36.8 | 56.8 | 69.4 | 96.9 | 44.6 | 63.7 |
| DARE-TIES | 81.4 | 74.5 | 50.8 | 39.2 | 55.0 | 70.7 | 97.6 | 40.1 | 63.7 |
| **KnOTS**-TIES | **82.7** | 73.7 | 49.3 | **48.9** | **68.9** | 70.9 | 95.5 | **53.8** | **68.0** |
| **KnOTS**-DARE-TIES | 81.8 | **75.9** | **50.7** | 40.3 | 53.2 | 70.2 | **97.9** | 41.0 | 63.9 |

all DARE experiments across three separate seeds, and report the best performance. KnOTS can be coupled with merging methods such as these latter three by applying them directly on aligned $[V^{(i)}]^T$ parameters. Note that we make an exception for methods like TIES (Yadav et al., 2023) that involve magnitude-based model pruning. While model-scale is absent in $[V^{(i)}]^T$ (its row-wise magnitudes are $\leq 1$ by definition), it is retained in $\Sigma$. Thus, we may apply such operations on $\Sigma[V^{(i)}]^T$, but merge using the newly pruned $[V^{(i)}]^T$s. Additionally, we observe that applying KnOTS on TA reduces to performing TA: $\sum_{i=1}^{n} \alpha^{(i)} \Delta W_j^{(i)} = \sum_{i=1}^{n} \alpha_i U \Sigma [V_j^{(i)}]^T = U \Sigma \sum_{i=1}^{n} \alpha_i [V_j^{(i)}]^T$. Thus, KnOTS on TA can be considered a *generalization* of TA: KnOTS reduces to the standard merging method when its alignment is not leveraged. We refer to all methods we apply KnOTS to by prefixing their names with KnOTS (e.g., KnOTS on TIES is denoted by KnOTS-TIES). Inspired by the recommendations made by each merging baseline, we tune hyperparameters such as scaling-coefficient, top-K and DARE-pruning-co-efficient as described in C. Note: KnOTS does not require any new hyperparameter to be tuned apart from the ones used by the original merging methods.

In some experiments, we add the "Ensemble" as a baseline, which consists of all models used in a particular merging evaluation. The ensemble passes an input in parallel to all models, and predicts its class according to the highest confidence prediction across all models.

**Metrics.** We report the *absolute* accuracy of all individual finetuned models on their respective datasets, and utilize these for merging. Similar to Ilharco et al. (2023); Yadav et al. (2023), we compare our merging methods via "normalized-accuracy" wherever applicable. This is obtained by dividing the performance of a merged model on a task (e.g., Cars (Krause et al., 2013)) by the performance of the original model finetuned on the task. For instance, the normalized accuracy for a given task-i is computed as $\frac{\text{Accuracy of merged model on task-i}}{\text{Accuracy of finetuned model on task-i}}$. This metric shows *how close* the merged model gets to the original finetuned model for each task. Certain experiment settings study different generalization properties and thus have different metrics. For the remainder of this paper, we assume all accuracy measurements are *normalized*, and define other metrics in their relevant sections.

## 5.2 PER-TASK EVALUATIONS ON VISION AND NLP SETTINGS

We first compare KnOTS's ability to improve existing merging methods in the popular "per-task" experiment settings (Ilharco et al., 2023; Yadav et al., 2023; Tang et al., 2024). These settings comprise of merging a set of models independently finetuned on different datasets into a single model, and then evaluating it on each dataset independently—using only its examples and labels. This way, the merged model is judged on its ability to preserve the individual skills of each original model.

**Merging eight ViT-B/32 models finetuned on image classification datasets.** We follow the image classification benchmark from Ilharco et al. (2023) and merge models finetuned on eight different datasets: Cars (Krause et al., 2013), DTD (Cimpoi et al., 2014), EuroSAT (Helber et al., 2019),

GTSRB Stallkamp et al. (2011), MNIST (LeCun, 1998), RESISC45 (Cheng et al., 2017), SUN397 (Xiao et al., 2016) and SVHN (Netzer et al., 2011). Like Ilharco et al. (2023); Yadav et al. (2023), we report normalized per-task accuracies and their average. Tab. 1 reports merging performances. Interestingly, nearly all merging methods achieve comparable performance. However, applying KnOTS on both TIES and DARE-TIES elevates their respective performances, with KnOTS-TIES strengthening TIES by 4.3% average normalized accuracy. Note, we also compare against a popular gradient-based approach—Fisher weight averaging (Matena & Raffel, 2022) in App. E and achieve considerable gains, though gradient-based merging is not our focus.

**KnOTS scales to ViT-L/14 models.** We also evaluate how KnOTS is affected by merging larger vision models. Specifically, we merge eight ViT-L/14 models finetuned with LoRA on the same datasets as our ViT-B/32 counterparts. We then evaluate the merged model's normalized per-task accuracies over each dataset. Results for several merging methods are summarized in Tab. 2.

Table 2: **Normalized per-task avg. image classification results.** We merge eight ViT-L/14 models finetuned with LoRA on eight vision datasets. We report the average normalized accuracies against average absolute accuracy of the finetuned models: 92.3%. KnOTS is best.

|  |  |  | KnOTS | |
|---|---|---|---|---|
| TA | TIES | DARE-TIES | TIES | DARE-TIES |
| 74.4 | 75.2 | 74.7 | **78.2** | 75.6 |

While performance of all merging methods improve with the larger models, KnOTS notably *consistently* outperforms the baselines in both KnOTS-TIES and KnOTS-DARE-TIES. Additionally, KnOTS-TIES is still capable of improving TIES by 3%, showing its ability to maintain performance with scale.

**KnOTS performs well on LLMs.** We also evaluate KnOTS in the NLI setting, by merging six PEFT llama3-8B (AI, 2024) models finetuned on SNLI (Bowman et al., 2015), MNLI (Williams et al., 2018), SICK (Marelli et al., 2014), QNLI, RTE (Wang et al., 2019), and SCITAIL (Khot et al., 2018). Each of these tasks involves performing a 3-way classification to determine whether a given "hypothesis" is true (entailment), false (contradiction) or inconclusive (neutral) compared to a given "premise." QNLI, RTE and SCITAIL

Table 3: **Normalized per-task avg. NLI results.** We merge six LLama3-8B models finetuned with LoRA on different NLI datasets. All numbers are normalized against the absolute average per-task accuracy of the individual finetuned models: 92.9%. KnOTS performs best.

|  |  |  | KnOTS | |
|---|---|---|---|---|
| TA | TIES | DARE-TIES | TIES | DARE-TIES |
| 90.2 | 90.0 | 90.9 | **92.9** | 91.1 |

only employ two of the three classes, so we simply mask the missing label when finetuning and evaluating on these datasets. Tab. 3 shows the results of merging these models using TA, TIEs, DARE-TIES, KnOTS-TIES and KnOTS-DARE-TIES. Overall, we observe that KnOTS-TIES *significantly* outperforms baseline merging methods by up to 2.9% average normalized accuracy, and KnOTS-DARE-TIES further improves DARE-TIES by .2%. This demonstrates the power of utilizing KnOTS to align LoRA models even on dramatically larger models (8B parameters), and its robustness across modalities.

## 5.3 A NEW BENCHMARK TOWARDS BUILDING GENERAL MODELS

We further introduce a new experimental setting to the eight vision per-task benchmark in Tab. 1, that we call the "joint-task." This task differs from the "per-task" regime in that it evaluates merged models over the union of all inputs and labels across *every* vision-task. The joint-task is *significantly* more challenging than the per-task counterpart as it explicitly examines whether a merged model is a *general* model: whether it can classify any input to any label.

Table 4: **Eight models joint-task results.** We merge eight ViT-B/32 models finetuned with LoRA on different datasets. We report the joint-task "Union" performances for several merging methods. KnOTS-TIES performs the best.

| Metric | Method | | | | | |
|---|---|---|---|---|---|---|
|  |  |  |  |  | KnOTS | |
|  | Ensemble | TA | TIES | DARE-TIES | TIES | DARE-TIES |
| Hits@1 | 40.7 | 43.6 | 43.6 | 44.0 | **46.8** | 45.2 |
| Hits@3 | 63.1 | 65.3 | 65.3 | 66.4 | **68.1** | 66.9 |
| Hits@5 | 72.6 | 74.0 | 73.9 | 75.1 | **76.3** | 75.3 |

After aggregating the labels across all eight tasks and removing duplicates (e.g., MNIST (LeCun, 1998) and SVHN (Netzer et al., 2011) have the same labels), we obtain 748 unique labels over which to classify all the images across all datasets. As some labels in one task are hyponyms of those in another (e.g., "islet" in SUN397 (Xiao et al., 2016) and

"island" in RESISC45 (Cheng et al., 2017)) making it challenging to distinguish labels, we report performance using "Hits@$k$." This refers to the number of times the expected label is within the top-$k$ predictions of a model (e.g., Hits@1 is accuracy). Tab. 4 shows the results over the "Union" of all images and labels across all tasks, when merging our ViT-B/32 LoRA models. Please see App. F for performances broken down by dataset. Overall, KnOTS-TIES significantly outperforms every baseline at all Hits@$k$ levels on the extremely challenging "Union" evaluation—by up to 3.2% on Hits@1. Interestingly, we also observe that the ensemble performs particularly poorly in the joint setting. We posit this is due to certain models making over-confidently incorrect predictions on data from tasks they are not finetuned on, a notable issue in ensemble models (Kardan et al., 2021). We argue that the joint-task is an important benchmark for assessing a merged model's generality, and we hope future work expands upon it.

## 5.4 Additional Analysis & Ablations

We also conduct three analysis experiments to understand different facets of KnOTS. We conduct our analysis with KnOTS-TIES as it consistently achieves the best performance across *all* settings.

**KnOTS scales better with the number of models.** Following Ilharco et al. (2023); Yadav et al. (2023), we evaluate the performance of KnOTS on the eight per-task vision benchmark with our ViT-B/32 models as we increase the number of tasks being merged. Fig. 3 illustrates the same measuring performance with average normalized accuracies of the merged model, where the dots represent the performance of a model evaluated *only* on the tasks included in the merge, while the bars indicate the 95% confidence intervals based on random subselection of 28 task combinations.

Notably, KnOTS-TIES achieves *significantly* better merging results compared to the baselines, with a performance gap that remains *consistently* > 4% (for >2 tasks being merged), underscoring its robustness as the number of tasks increases.

**KnOTS is robust to different LoRA ranks.** Fig. 4 shows the performance of KnOTS-TIES against TIES (Yadav et al., 2023) in merging our ViT-B/32 models trained at various LoRA ranks {4, 16, 64, 256, 768} on the eight per-task vision benchmark. Note that these ViTs each have a feature dimension of 768, making our LoRA rank 768 setting "*full rank*". The results consistently demonstrate that KnOTS-TIES outperforms TIES across all rank settings. In the very low rank setting (e.g., rank four), KnOTS improves TIES by 6% (64.6% normalized accuracy vs. 58.6%). The performance uplift continues with increasing LoRA rank, with KnOTS improving TIES by 4% in the full rank setting. This experiment highlights how KnOTS is both robust and scalable to merging models finetuned on diverse tasks with arbitrary LoRA rank.

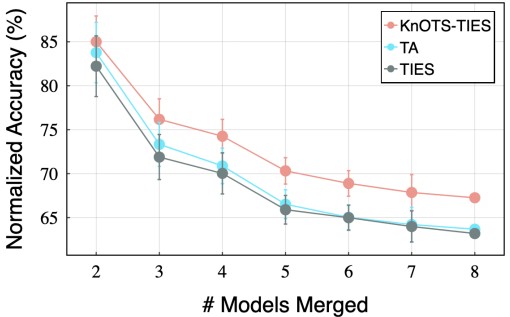

Figure 3: **KnOTS boosts performance with scale.** KnOTS-TIES continues to see gains, outperforming original TIES (Yadav et al., 2023) and Task Arithmetic (TA) (Ilharco et al., 2023) when merging an increasing number of tasks in the per-task evaluation vision setting § 5.2. Performance is the average normalized accuracy with 95% confidence intervals over merging different combinations of the tasks

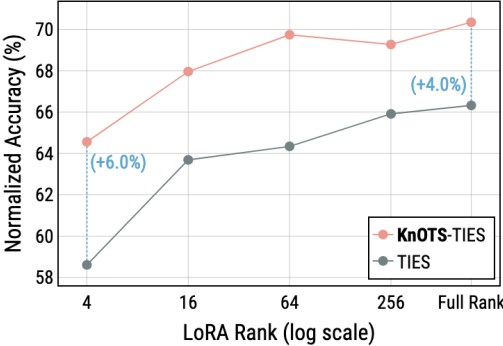

Figure 4: **KnOTS boosts performance across LoRA ranks.** KnOTS-TIES consistently outperforms TIES (Yadav et al., 2023) across varying LoRA ranks in the per-task evaluation vision setting. Performance is reported in terms of average normalized accuracy across eight vision tasks.

**The way task-updates are concatenated matters.** KnOTS concatenates task-updates column-wise before computing the SVD. In this experiment, we investigate the effects of concatenating them

row-wise instead. We conduct our analysis using the same vision models from Tab. 1 and merge them with KnOTS-TIES. We observe that applying KnOTS-TIES after concatenating the rows of the task-updates achieves an average normalized accuracy of 65.4%. It performs 2.6% worse than KnOTS-TIES with column-wise task-update concatenation. We posit this may be due to the difference in the deconstructed task-update terms when concatenating them row-wise. Specifically, each update would be denoted by $\Delta W_j^{(i)} = U^{(i)} \Sigma V$, where $\Sigma V$ are shared across all updates with different $U^{(i)}$. In this way, the shared $\Sigma V$ would act on distinct $U^{(i)}$ containing different information—which in turn may lower alignment. These results suggest that concatenating the updates column-wise is crucial to obtaining strong performance with KnOTS.

## 6 CONCLUSION

In this paper, we study merging LoRA models sharing the same pretrained checkpoint and finetuned on different tasks without additional finetuning. We find that prior work does not transfer well in this setting, and observe that this is due to parameter misalignment between LoRA models. We introduce KnOTS to tackle this problem by using the singular value decomposition (SVD) to map the parameters of each LoRA model into a shared representation space with aligned elements, over which prior work can be applied to merge the parameters. In addition, we introduce a novel benchmark designed to evaluate whether merged models can be *general*. Notably, KnOTS improves prior work by up to 4.3% across both vision and language settings, including our new setting.

## 7 REPRODUCIBILITY STATEMENT

We describe the procedure for finetuning all our models, including compute resources, datasets, and hyperparameter searches in App. D. App. C describes our hyperparameter searches for all merging methods. App. B explains how we calculated our centered kernel alignment (CKA) visualizations in Fig. 2. In addition, App. F provides extensive details on how we created our novel benchmark.

## 8 ACKNOWLEDGEMENTS

This work was partially sponsored by NSF awards #2403297 and #2144194, an NSF-GRFP, and Google. All views and conclusions expressed in this work are those of the authors and not a reflection of these sources.

We would like to thank Simar Kareer for his valuable insights and discussions that contributed to the brainstorming process of this work.

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

## A  WHICH LoRA REPRESENTATION IS BEST FOR MERGING?

LoRA is a popular Parameter Efficient Finetuning (PEFT) method that finetunes models by applying low-rank updates to the pretrained parameters. For instance, let $W_j^{(i)}, \Delta W_j^{(i)} \in \mathbb{R}^{O \times I}$. With LoRA, each $\Delta W_j^{(i)}$ is rank $r << \min(O, I)$, and can be factorized into two low-rank matrices $A_j^{(i)} \in \mathbb{R}^{r \times I}, B_j^{(i)} \in \mathbb{R}^{O \times r}$, with $\Delta W_j^{(i)} = B_j^{(i)} A_j^{(i)}$. The existence of $A_j^{(i)}$, and $B_j^{(i)}$ for each $\Delta W_j^{(i)}$ may suggest that merging should be done on them separately, and the results multiplied to achieve the merged-update $\Delta W_j^{(m)}$. However, this immediately leads to issues when applying existing merging methods. For instance, let us apply task-arithmetic (TA) Ilharco et al. (2023) on a single arbitrary layer (e.g., the $j^{th}$ layer) across $f^{(1)}, \dots, f^{(n)}$. First, TA merges the factorizations of $\Delta W_j^{(1)}, \dots, \Delta W_j^{(n)}$ by computing $\sum_{i=1}^{n} \lambda^{(i)} A_j^{(i)}, \sum_{i=1}^{n} \lambda^{(i)} B_j^{(i)}$. Multiplying each to obtain $\Delta W_j^{(m)}$ yields,

$$\Delta W_j^{(m)} = \left( \sum_{i=1}^{n} \lambda^{(i)} B_j^{(i)} \right) \left( \sum_{i=1}^{n} \lambda^{(i)} A_j^{(i)} \right) \qquad (3)$$

$$= \underbrace{\left( \sum_{i=1}^{l} \lambda^{(i)} \lambda^{(i)} B_j^{(i)} A_j^{(i)} \right)}_{\text{Products of aligned factorizations}} + \underbrace{\left( \sum_{i \neq k}^{l} \lambda^{(i)} \lambda^{(k)} B_j^{(i)} A_j^{(k)} \right)}_{\text{Products of misaligned factorizations.}} . \qquad (4)$$

Notably, $\Delta W_j^{(m)}$ consists of two terms: one composed of $B_j^{(i)}$ and $A_j^{(i)}$ matrices from the updates of the same model, and one composed of $B_j^{(i)}$ and $A_j^{(k)}$ matrices from updates of different models. Unfortunately, there is no guarantee that $B_j^{(i)}$ expects the same input representation as the output of $A_j^{(k)}$ when $i \neq k$, and including these terms in the merged model can incur significant drop in model performance Stoica et al. (2024). Similarly, this same issue persists as well when applying Yadav et al. (2023). Thus, we only conduct merging on the original update representations: $\Delta W_j^{(i)}$ of each model. This trivially avoids mismatches when applying merging methods.

## B  COMPUTING CENTERED KERNEL ALIGNMENTS ACROSS TASKS

We create the pairwise centered kernel alignment (CKA) (Kornblith et al., 2019) matrices presented in Fig. 2 using models that are finetuned from the same pretrained checkpoint on the different tasks shown in Tab. 1. Each matrix is generated using a set of unlabeled heldout data from *all* eight datasets. Specifically, this heldout set consists of the validation data of the respective dataset when it exists and otherwise randomly samples 20% of the test set. Note that in situations where we sample 20% of a dataset's test split, we always evaluate any merged model on the remaining 80% of examples. For fair comparison, all CKA plots are created using the *same* data.

**Pairwise CKA on full-rank finetuned models.** We compute Fig. 2a) as follows. We first take the set of eight full-rank finetuned (FFT) ViT-B/32 (Dosovitskiy et al., 2021) models released by Ilharco et al. (2023), along with their shared pretrained model. We then collect the intermediate outputs of every key, query, value and projection layer across all nine models over all the data from our heldout set. This yields nine different intermediate activations for each layer across all models. We then subtract the activations of the pretrained model from the eight activations of the finetuned models, at every layer. We then compute the CKA (following (Kornblith et al., 2019)) between activations of two different models at every layer, across all model pairs. Finally, we average the results over all layers.

**Pairwise CKA on LoRA finetuned models.** We compute Fig. 2b) as follows. We first take the set of eight LoRA models we merge in in Tab. 1, and extract the intermediate activations over all LoRA layers of each model and over the same data used in our FFT CKA plots. Specifically, these layers are every key, query, value and projection layer of each model. Since LoRA finetuning *adds* new parameters to a pretrained model, there are no layers from the pretrained model to subtract. Thus, we

directly compute CKA over the activations at the same layer across LoRA model pairs, and average over all layers.

**Pairwise CKA on LoRA finetuned models aligned with KnOTS.** We compute Fig. 2c) as follows. This procedure is extremely similar to computing the CKAs for LoRA finetuned models, with one small difference. Specifically, rather than computing the CKAs from the output activations of LoRA layers, we instead calculate the CKA over the outputs of different $[V^{(i)}]^T$ for the same layer of different LoRA models (using the same data again). Afterwards, we average the results across all layers.

## C    TUNING THE HYPERPARAMETERS FOR DIFFERENT MERGING BASELINES

Scaling co-efficient: A single scalar scaling co-efficient is tuned across the range [0.1, 0.2, 0.3,..., 1.0]

top-$k$ : Like Yadav et al. (2023), we define top-$k$ the percentage of task-update elements retained when merging. It is tuned across the range [10, 20, 30,..., 100]

DARE-pruning co-efficient ($p$): [0.99, 0.9, 0.8, ..., 0.1]

DARE-Seeds: We evaluated DARE over the following five seeds: [420, 421, 422, 423, 424].

For methods such as TIES (Yadav et al., 2023) and DARE (Yu et al., 2024) which involve tuning more than one scaling co-efficient we tune by performing a linear-search by first tuning the scaling co-efficient and then the corresponding pruning hyperparameter i.e top-$k$ for TIES and DARE-pruning co-efficient for DARE. The default value used for TIES top-$k$ is 30 and for the DARE-pruning co-efficient is 0.9 as recommended by their respective baselines.

## D    TRAINING DETAILS

**Training Vision models.** We make use of the CLIP (Radford et al., 2021) based ViT-B/32 and ViT-L/14 models from Hugging Face (HF). These models are then LoRA fine-tuned using HF's PEFT LoRA library. Across all our experiments we initialize the LoRA layers across the query, key, value and output layer. Note that these are the only learnable layers. We set the LoRA rank to be 16, LoRA alpha to be 16, LoRA dropout to be 0.1 and disable the use of bias parameters. All models are trained using the AdamW (Loshchilov & Hutter, 2019) optimizer, with a cosine learning rate scheduler (Loshchilov & Hutter, 2017) using Cross-Entropy loss.

The ViT-B/32 models were fine-tuned on the 8 vision tasks using a standard learning rate of 1e-5, weight decay of 1e-1 and label smoothing set to 0.

The ViT-L/14 models were fine-tuned on the 8 vision tasks using a standard learning rate of 3e-4, weight decay of 1e-4 and label smoothing set to 0.

Across all our experiments the text encoder in the CLIP-model remains frozen and the text embeddings are obtained by passing the class labels through the text encoder.

**Training LLMs on NLI tasks.** For the setting in § 5.2 we use the LLama-3 8B parameter model. We initialize the model using the "AutoModelForSequenceClassification" model class from Hugging Face's "AutoModel" to classify across 3 classes. The models are LoRA fine-tuned using the PEFT LoRA library. We initialize rank-16 LoRA weights across the weights of the attention layer only i.e the query, key, value and output layer. The models were trained using AdamW (Loshchilov & Hutter, 2019) optimizer using a linear learning rate scheduler, with a learning rate of 3e-5 and warm steps set to 6% of the total number of training steps. For datasets like QNLI (Wang et al., 2019), RTE (Wang et al., 2019) and SCITAIL (Khot et al., 2018) which only employ two of the three classes, we simply mask the missing label when finetuning and evaluating on these datasets.

During merging we only merge the Llama back-bone with the LoRA layers and use task-specific head when evaluating across each dataset. Even for the merged model we use the merged backbone and task-specific head during evaluation.

Table A1: **Eight models per-task results including Fisher weight averaging.** We merge eight ViT-B/32 models finetuned with LoRA on different image classification datasets. "Finetuned" refers to the accuracy of each finetuned model on the dataset it was trained on. We report the per-task (including the average) normalized-accuracies across other merging baselines. These describe how close they get to the "Finetuned" accuracy.

| Method | Datasets | | | | | | | | |
|---|---|---|---|---|---|---|---|---|---|
| | Cars | DTD | EuroSAT | GTSRB | MNIST | RESISC45 | SUN397 | SVHN | Avg |
| | Per-Task Absolute Accuracies (%) | | | | | | | | |
| Finetuned | 74.0 | 58.3 | 99.0 | 92.7 | 99.3 | 88.4 | 64.5 | 96.2 | 84.1 |
| | Per-Task Accuracies of Merged Models Normalized Against Finetuned Models (%) | | | | | | | | |
| RegMean | 80.2 | 71.3 | 37.9 | 47.3 | 43.1 | 70.5 | 93.9 | 43.0 | 60.9 |
| Fisher | 84.5 | 72.4 | 44.4 | **55.8** | 47.8 | 70.9 | 96.1 | 39.2 | 63.9 |
| TA | 82.0 | 73.6 | 48.8 | 42.1 | 53.1 | **71.5** | 97.5 | 41.2 | 63.7 |
| TIES | 82.2 | 72.8 | 50.0 | 36.8 | 56.8 | 69.4 | 96.9 | 44.6 | 63.7 |
| DARE-TIES | 81.4 | 74.5 | 50.8 | 39.2 | 55.0 | 70.7 | 97.6 | 40.1 | 63.7 |
| **KnOTS**-TIES | **82.7** | 73.7 | 49.3 | 48.9 | **68.9** | 70.9 | 95.5 | **53.8** | **68.0** |
| **KnOTS**-DARE-TIES | 81.8 | **75.9** | **50.7** | 40.3 | 53.2 | 70.2 | **97.9** | 41.0 | 63.9 |

**Compute resources.** All of our experiments were conducted on machines with one Nvidia A40 with 48GB of VRAM, and a CPU that has 8 workers. We trained all our models across these machines, and also applied every merging algorithm in this environment. KnOTS is capable of running entirely on the CPU. We compute the SVD using the Pytorch (Paszke et al., 2019) "torch.linalg.svd" solver. However, we note that more efficient SVD algorithms can easily be employed with our approach, such as the recent Fast SVD (Xu et al., 2023) which is designed for low-rank matrices.

**Datasets and licenses.** This paper uses the following datasets and associated licenses. Cars (Krause et al., 2013) and GTSRB (Stallkamp et al., 2011) both use the Creative Commons License. EuroSAT (Helber et al., 2019) is under the MIT license and MNIST (LeCun, 1998) is under the Gnu General Public License. We could not find the licenses of DTD (Cimpoi et al., 2014), RESISC45 (Cheng et al., 2017), SVHN (Netzer et al., 2011) and SUN397 (Xiao et al., 2016).

## E    COMPARISON WITH A FINETUNING BENCHMARK

We also compare KnOTS against the popular finetuning benchmark Fisher Weight Averaging (Matena & Raffel, 2022) in the same setting as Tab. 1. We classify this method as finetuning because it adds learnable parameters to every model that are optimized via gradient descent to obtain the best merged model. We select the best reported Fisher model according to the hyperparameter configuration which achieved the best performance on the same held-out validation data as all methods in this setting. Our hyperparameter search consisted of two ranges. First, the number of examples used to compute the Fisher information matrices: [256, 512, 1024, 2048] (selected from the training data of each dataset). Second, the scaling term to merge model parameters: [0.1, 0.2, 0.3, 0.4, 0.5, 0.6, 0.7, 0.8, 0.9]. The best performing configuration on our held-out validation dataset used a scaling coefficient of 1.0 and required 256 examples to compute the Fisher weights. Tab. A1 summarizes the results. Overall, we observe that while Fisher matches TIES, it is substantially outperformed by KnOTS—a method that doesn't require any training.

## F    JOINT TASK FULL PERFORMANCES

Tab. A2 shows performances for each merging method across all datasets on the Joint-task. Each dataset column shows results on images *only* from the dataset, but with using the *joint* labels-set of 748 labels.

**Finding synonymous labels.** After removing duplicates from the joint label space, we are left with 748 total labels. However, a manual search over these labels reveals that some may be synonyms/hyponyms of each other. To get a better idea of how many there are, we encode each label using a

Table A2: **Eight models joint-task results.** We merge eight ViT-B/32 models finetuned with LoRA on different datasets. We report the joint-task performances for several merging methods. KnOTS-TIES improves over baselines by significant margins across across nearly every evaluation, often only trading best performances with the "Ensemble."

| Method | Metric | Joint-Task Performances (%) | | | | | | | | |
|---|---|---|---|---|---|---|---|---|---|---|
| | | Cars | DTD | EuroSAT | GTSRB | MNIST | RESISC45 | SUN397 | SVHN | Union |
| Ensemble | Hits@1 | 58.5 | **41.3** | **16.9** | 29.5 | 35.8 | 54.2 | **62.4** | 25.1 | 40.7 |
| | Hits@3 | 83.6 | 61.4 | **31.4** | 59.5 | **58.0** | 78.2 | **84.0** | 44.3 | 63.1 |
| | Hits@5 | 91.0 | 72.4 | **40.0** | 73.0 | **69.1** | 85.9 | 89.5 | 55.2 | 72.6 |
| TA | Hits@1 | 60.7 | 40.7 | 15.3 | 38.8 | 31.8 | **59.7** | 61.9 | 29.2 | 43.5 |
| | Hits@3 | 84.9 | 63.7 | 23.0 | 66.1 | 48.4 | 83.6 | 83.9 | 50.1 | 65.2 |
| | Hits@5 | 92.0 | 74.0 | 31.0 | 77.9 | 55.7 | 90.2 | **89.9** | 61.6 | 74.0 |
| TIES | Hits@1 | 60.4 | 39.7 | 13.0 | 35.0 | 33.4 | 58.6 | 61.3 | 32.9 | 43.6 |
| | Hits@3 | 84.9 | 61.9 | 21.9 | 63.3 | 48.2 | 82.8 | 83.7 | 53.6 | 65.3 |
| | Hits@5 | 92.1 | 72.4 | 29.0 | 75.3 | 54.0 | 89.4 | 89.9 | 64.1 | 73.9 |
| DARE-TIES | Hits@1 | 60.8 | 39.3 | 12.8 | 33.7 | 34.3 | 57.5 | 60.4 | 35.5 | 44.0 |
| | Hits@3 | 85.3 | 61.4 | 18.1 | 63.6 | 50.2 | 82.3 | 82.6 | 57.8 | 66.4 |
| | Hits@5 | 92.6 | 73.0 | 20.9 | 74.6 | 55.7 | 89.0 | 89.1 | 69.5 | 75.1 |
| **KnOTS**-TIES | Hits@1 | **61.7** | 40.5 | 16.2 | **44.2** | **39.1** | 59.0 | 60.6 | **36.7** | **46.8** |
| | Hits@3 | **85.8** | **63.8** | 22.3 | **69.0** | 52.6 | **83.9** | 82.8 | **58.4** | **68.1** |
| | Hits@5 | **92.6** | **74.5** | 31.1 | **79.8** | 58.4 | **90.4** | 89.1 | **68.5** | **76.3** |
| **KnOTS**-DARE-TIES | Hits@1 | 60.4 | 40.3 | 15.9 | 41.9 | 34.6 | 58.4 | 60.4 | 34.8 | 45.2 |
| | Hits@3 | 85.0 | 63.5 | 21.1 | 68.4 | 50.0 | 83.8 | 82.4 | 56.5 | 66.9 |
| | Hits@5 | 92.2 | 74.5 | 26.6 | 78.9 | 55.9 | **90.4** | 88.9 | 67.4 | 75.3 |

pretrained "distilbert-base-nli-mean-tokens" model taken from the SentenceTransformers library (Reimers & Gurevych, 2019), and compute the pairwise cosine-similarities between each label. We then take all label pairs with cosine-similarity $\geq 0.8$ to be "synonyms," and reproduce them at the bottom of this section in dictionary form where keys and values are tuples containing "(label, dataset origin)." In total, we find 111 synonyms, leading to an average of $0.15$ synonyms *per-label* and $1.95$ synonyms *per-labels that have synonyms*. Thus, measuring performance on the joint-task using "Hits@k" with $k = \{1, 3, 5\}$ appears to be a suitable choice.

Below we print all synonyms found by our automatic search.

```
found_synonyms = {
('lake natural', 'sun397'): [('lake', 'resisc45')],
('forest', 'resisc45'): [
    ('tree house', 'sun397'), ('rainforest', 'sun397'),
    ('forest broadleaf', 'sun397'), ('forest road', 'sun397'),
    ('forest needleleaf', 'sun397'), ('forest path', 'sun397')
],
('athletic field outdoor', 'sun397'): [('ground track field', 'resisc45')],
('lake or sea', 'eurosat'): [('lake', 'resisc45')],
('rainforest', 'sun397'): [('forest', 'resisc45')],
('iceberg', 'sun397'): [('snowberg', 'resisc45')],
('meadow', 'resisc45'): [
    ('field wild', 'sun397'), ('park', 'sun397'), ('field cultivated', 'sun397'),
    ('pasture land', 'eurosat'), ('yard', 'sun397'), ('pasture', 'sun397')
],
('pond', 'sun397'): [('lake', 'resisc45')],
('ground track field', 'resisc45'): [
    ('athletic field outdoor', 'sun397'), ('track outdoor', 'sun397')
],
('desert vegetation', 'sun397'): [('desert', 'resisc45')],
('marsh', 'sun397'): [('wetland', 'resisc45')],
('freeway', 'resisc45'): [('highway', 'sun397')],
('islet', 'sun397'): [('island', 'resisc45')],
('permanent crop land', 'eurosat'): [
    ('rectangular farmland', 'resisc45'), ('field cultivated', 'sun397')
],
('track outdoor', 'sun397'): [('ground track field', 'resisc45')],
('swamp', 'sun397'): [('wetland', 'resisc45')],
```

```
('sea ice', 'resisc45'): [('ice shelf', 'sun397'), ('ice floe', 'sun397')],
('desert', 'resisc45'): [
    ('desert vegetation', 'sun397'), ('desert sand', 'sun397')
],
('snowberg', 'resisc45'): [
    ('iceberg', 'sun397'), ('mountain snowy', 'sun397'), ('snowfield', 'sun397')
],
('railway station', 'resisc45'): [
('train railway', 'sun397'), ('railroad track', 'sun397'), (
'train station platform', 'sun397')
],
('street', 'sun397'): [
    ('commercial area', 'resisc45'), ('intersection', 'resisc45')
],
('rectangular farmland', 'resisc45'): [('permanent crop land', 'eurosat')],
('field wild', 'sun397'): [('meadow', 'resisc45')],
('thermal power station', 'resisc45'): [('electrical substation', 'sun397')],
('lake', 'resisc45'): [
    ('lake natural', 'sun397'), ('lake or sea', 'eurosat'), ('pond', 'sun397')
],
('runway', 'sun397'): [('airplane', 'resisc45'), ('airport', 'resisc45')],
('baseball field', 'sun397'): [('baseball diamond', 'resisc45')],
('industrial area', 'sun397'): [
    ('industrial buildings or commercial buildings', 'eurosat')
],
('ice shelf', 'sun397'): [('sea ice', 'resisc45')],
('airplane', 'resisc45'): [('runway', 'sun397'), ('airplane cabin', 'sun397')],
('park', 'sun397'): [('forest', 'resisc45'), ('meadow', 'resisc45')],
('thermal power station', 'resisc45'): [('electrical substation', 'sun397')],
('electrical substation', 'sun397'): [('thermal power station', 'resisc45')],
('field cultivated', 'sun397'): [
    ('meadow', 'resisc45'), ('permanent crop land', 'eurosat'),
    ('circular farmland', 'resisc45'), ('pasture land', 'eurosat')
],
('patio', 'sun397'): [('terrace', 'resisc45')],
('shopfront', 'sun397'): [('commercial area', 'resisc45')],
('highway', 'sun397'): [('freeway', 'resisc45'), ('highway or road', 'eurosat')],
('circular farmland', 'resisc45'): [
    ('field cultivated', 'sun397'), ('pasture land', 'eurosat'),
    ('pasture', 'sun397')
],
('residential buildings or homes or apartments', 'eurosat'): [
    ('residential neighborhood', 'sun397')
],
('pasture land', 'eurosat'): [
    ('meadow', 'resisc45'), ('field cultivated', 'sun397'),
    ('circular farmland', 'resisc45'), ('yard', 'sun397'),
    ('pasture', 'sun397')
],
('desert sand', 'sun397'): [('desert', 'resisc45')],
('train railway', 'sun397'): [('railway', 'resisc45')],
('wetland', 'resisc45'): [('marsh', 'sun397'), ('swamp', 'sun397')],
('commercial area', 'resisc45'): [
    ('street', 'sun397'), ('shopfront', 'sun397'),
    ('industrial buildings or commercial buildings', 'eurosat')
],
('industrial buildings or commercial buildings', 'eurosat'): [
    ('industrial area', 'sun397'), ('commercial area', 'resisc45')
],
```

```
('airport', 'resisc45'): [('airport terminal', 'sun397'), ('runway', 'sun397')],
('tennis court outdoor', 'sun397'): [('tennis court', 'resisc45')],
('baseball diamond', 'resisc45'): [
    ('baseball field', 'sun397'), ('stadium baseball', 'sun397'),
    ('batters box', 'sun397')
],
('railway', 'resisc45'): [
    ('train railway', 'sun397'), ('railroad track', 'sun397'),
    ('train station platform', 'sun397')
],
('highway or road', 'eurosat'): [('highway', 'sun397')],
('yard', 'sun397'): [('meadow', 'resisc45'), ('pasture land', 'eurosat')],
('railroad track', 'sun397'): [
('railway station', 'resisc45'), ('railway', 'resisc45')
],
('tennis court', 'resisc45'): [
    ('tennis court outdoor', 'sun397'), ('tennis court indoor', 'sun397')
],
('residential neighborhood', 'sun397'): [
    ('residential buildings or homes or apartments', 'eurosat')
],
('island', 'resisc45'): [('islet', 'sun397')],
('train station platform', 'sun397'): [('railway station', 'resisc45')],
('pasture', 'sun397'): [
    ('meadow', 'resisc45'), ('circular farmland', 'resisc45'),
    ('pasture land', 'eurosat')
],
('terrace', 'resisc45'): [
    ('courtyard', 'sun397'), ('promenade deck', 'sun397'),
    ('pavilion', 'sun397'), ('patio', 'sun397'),
    ('carrousel', 'sun397'), ('balcony exterior', 'sun397'),
    ('veranda', 'sun397')
],
}
```

