# OpenReview forum: "Model merging with SVD to tie the Knots"
_ICLR.cc/2025/Conference — ICLR 2025 Poster_

### Official Review · Reviewer_6fFE · 2024-10-27

**Soundness:** 3
**Presentation:** 3
**Contribution:** 2
**Rating:** 6
**Confidence:** 4

**Summary:**

This paper introduces a method called KnoTs which provides a strategy for merging LoRA modules. First the authors show that centered kernel alignment (CKA) between activations of LoRA modules is not as high as that of fully fine-tuned models, demonstrating the need for developing methods specially for merging LoRA modules. They propose to perform SVD on the task updates and perform merging operation on V matrix in the SVD which is known to contain task specific information. First they show that CKA on these V matrix is high and this would eventually help in improving merged model performance. They run experiments on both vision and language tasks to demonstrate the effectiveness of the method and their method achieves better normalized accuracy in comparison to other SOTA methods specially designed for merging fully fine-tuned models.

**Strengths:**

* CKA analysis provided in the beginning to show misalignment of weights in LoRA provides good motivation to develop specialized method for merging LoRA trained models.
* Proposes a merging method for combining LoRA modules using SVD decomposition on weight update matrix. Shows improvements over different merging baselines for both vision and language tasks.
* KnoTs demonstrates superior scalability compared to other methods, particularly as the number of merged models increases.

**Weaknesses:**

* This method doesn't allow merging of LoRA modules with different ranks.
* The datasets used in this work are selected to ensure strong baseline performance, where models like ViT and Llama already demonstrate high zero-shot accuracy. This high initial performance makes it challenging to quantify the specific gains achieved through the proposed method.
* Multi-task trained performance should be considered as one of the baseline to understand how far is the merging performance from multi-task performance.

**Questions:**

* What is the motivation for performing merging operation on V matrix from SVD?
* Does the author believe that merging methods would have a greater negative impact if the base model’s zero-shot performance were lower?
* Is it possible to compare merged models performance to multi-task model performance on all the datasets?

---

> ### Author Response · Authors · 2024-11-23
> **Author response 1/2 to reviewer 6fPE**
>
> > **Weakness 1**: This method doesn't allow merging of LoRA modules with different ranks.
>
> We are unsure of what the reviewer means by “different ranks.” Specifically, is the reviewer asking whether KnOTS can merge LoRA models in which each has a different rank? Or if the reviewer is asking if KnOTS is robust to merging LoRA models beyond rank 16? We have prepared responses to both questions below. However, we ask the reviewer to let us know if they were asking a different question.
>
> *Is KnOTS robust to merging models beyond rank 16?* \
> We have updated our submission PDF to include an experiment (see Fig. 4 in Section 5.4) that examines KnOTS’s capability of obtaining strong merged models across varying LoRA ranks in the eight per-task vision benchmark described in Section 5.2. We conduct our analysis by first choosing a rank from {4,16,64,256,768}, and then LoRA finetuning all our ViT-B/32 models with this rank. We compare the performance of our best performing baseline (TIES) and our best performing method (KnOTS-TIES) as LoRA rank increases. We observe that KnOTS significantly improves baseline merging performance across all ranks, ranging from very small (e.g., 4) to very large (e.g., full rank), highlighting its robustness and scalability.
>
> *Is KnOTS capable of merging models with different individual ranks?* \
> This setting is a trivial extension of KnOTS, and we prove this here. Consider that we wish to merge $n \geq 2$  LoRA models, each finetuned from the same pretrained model with (different) rank $r_i\in\mathbb{N}$ ($i = \{1,2,\ldots,n\}$). Let their respective task-updates be represented by $\{\Delta W^{(1)}\ldots, \Delta W^{(n)}\}\in\mathbb{R}^{O\times I}$ for notation consistency. Just as presented in Section 4, KnOTS transforms these into, $U\Sigma \left[V^{(1)},\ldots, V^{(n)}\right]$, where $U\in\mathbb{R}^{O\times k}$ ($k \leq \min\left(I, O, \sum_{i=1}^{n} r_{i}\right)$), $\Sigma\in\mathbb{R}^{k\times k}$ and $\{V^{(1)},\ldots, V^{(n)}\}\in\mathbb{R}^{k\times I}$. Merging approaches can then be applied without change.
>
> > **Weakness 2**: The datasets used in this work are selected to ensure strong baseline performance, where models like ViT and Llama already demonstrate high zero-shot accuracy. This high initial performance makes it challenging to quantify the specific gains achieved through the proposed method.
>
> We clarify that nearly all of our ViT-based vision experiments strictly adhere to the established benchmark proposed by [1], which to the best of our knowledge has become the de facto standard used by prior published works (appearing in ICLR and NeurIPS) that evaluate on vision [1, 2, 3, 4]. While it is true that Llama is a very powerful model, the principal motivation behind our language setting selection is to better understand how KnOTS scales to dramatically larger models and how it may behave in language applications. Despite the representation capabilities of Llama, existing published baselines are unable to achieve superior merges in our NLI setting. Instead, KnOTS is still capable of significantly outperforming them by nearly 3%, highlighting its strength as an alignment method across even very large language models. As the reviewer mentions, improving where the model’s performance on a task is high is even harder and hence signifies a more meaningful improvement.
>
> We do share the reviewer’s perspective on the importance of comparing alignment/merging methods in challenging environments. This motivated us to introduce the new joint variant of the vision benchmark proposed by [1] (Section 5.3). This setting unifies all the datasets from the benchmark into a single collection, where each image must be correctly classified from the labels of all datasets together. This setting is significantly more challenging because merged models are no longer given privileged information from the task they are evaluated on (i.e., discriminating only amongst the labels pertaining to the task). Thus, we argue that the “joint setting” examines a model’s ability at being “general.”
> Note that in our gradient-free setting, evaluating the capabilities of individual base models relies on their ensemble. KnOTS-TIES demonstrates an up to 2.9% improvement over TIES, and a large 6.4% improvement over this ensemble. Notably, KnOTS-TIES establishes superior performance despite its merged model never being trained on this setting.
>
> Overall KnOTS is consistently better than baselines across all settings evaluated, highlighting its ability to improve the strength of merging methods.
>
> *References*: \
> [1] Ilharco et al., Editing models with task arithmetic. (ICLR 2023) \
> [2] Yadav, et al., Ties-merging: resolving interference when merging models. (NeurIPS 2023) \
> [3] Ortiz-Jimenez et al., Task arithmetic in the tangent space: Improved editing of pre-trained models. (NeurIPS 2023) \
> [4] Tang et al., Parameter-efficient multi-task model fusion with partial linearization. (ICLR 2024)

---

> ### Author Response · Authors · 2024-11-23
> **Author response 2/2 to reviewer 6fPE**
>
> > **Weakness 3**: Multi-task trained performance should be considered as one of the baseline to understand how far is the merging performance from multi-task performance.
>
> The performance of a multi-task model serves as a valuable upper bound for all model merging baselines, as it assumes privileged access to training data and gradients across all tasks, allowing it to resolve task interference during training. For instance, we find that the Llama3-8B LoRA multitask model finetuned in our NLI setting achieves 91.9% average accuracy, nearly matching the individual finetuned models' average accuracy of 92.5%. Our scope specifically examines gradient-free merging methods. These operate in the post-training regime, where task-specific models are trained individually with the same initialization. Thus, we argue that the multitask model is out of our scope, and performance of each individual base-model is the ultimate benchmark. However, we would expect the multitask performance to be very close to the individual finetuned performance across all remaining settings.
>
>
> > **Question 1**: What is the motivation for performing merging operation on V matrix from SVD?
>
> Figure 2 summarizes our motivation. We observe that fully finetuned models (over which existing merging methods work well) have very structurally similar task-updates (Fig 2a). On the other hand, LoRA finetuned models have considerably lower structural alignments (Fig 2b), yielding poorer merges. However, Fig 2c illustrates how the same LoRA task-updates are significantly better aligned in the V-space of the SVD. Based on this, we posit that merging models in this better-aligned V-space enables improved merging.
>
> > **Question 2**: Does the author believe that merging methods would have a greater negative impact if the base model’s zero-shot performance were lower?
>
> Yes and no. Initialization plays an important role in merging success [1,2,3,4,5,6,7,8,9,10]. Prior work has shown that models finetuned from strong pretrained models can be merged well [1,2,8,10], and argues that success is in part dependent on small finetuning updates (i.e., finetuning does not significantly change the underlying representation structure from the pretrained model) [1,2,10]. From this perspective, it may be hypothesized that models finetuned from poorer pretrained models are more challenging to merge. However, we note that pretrained capability is not the sole factor contributing to merging success. Strong merges also rely on strong base-models [1,2,6,9,10]. To this end, it is the opinion of the authors that models involved in merging should be experts in their respective settings to achieve strong success with merging. Thus, we may similarly expect the performance of the merging methods used in our work to improve as base-model capabilities improve from strong pretrained models.
>
> > **Question 3**: Is it possible to compare merged models performance to multi-task model performance on all the datasets?
>
> Please see our response to “Weakness 3”.
>
> *References*: \
> [1] Ilharco et al., Editing models with task arithmetic. (ICLR 2023) \
> [2] Yadav, et al., Ties-merging: resolving interference when merging models. (NeurIPS 2023) \
> [3] Ortiz-Jimenez et al., Task arithmetic in the tangent space: Improved editing of pre-trained models. (NeurIPS 2023) \
> [4] Tang et al., Parameter-efficient multi-task model fusion with partial linearization. (ICLR 2024) \
> [5] Entezari et al., The Role of Permutation Invariance in Linear Mode Connectivity of Neural Networks (ICLR 2022) \
> [6] Simsek et al., Geometry of the loss landscape in overparameterized neural networks: Symmetries and invariances. (PMLR 2021) \
> [7] Wortsman et al., Model soups: averaging weights of multiple fine-tuned models improves accuracy without increasing inference time (ICML 2022) \
> [8] Matena et al., Merging Models with Fisher-Weighted Averaging. (NeurIPS 2022) \
> [9] Stoica & Bolya et al., ZipIt! Merging Models from Different Tasks without Training (ICLR 2024) \
> [10] Yu et al., Language Models are Super Mario: Absorbing Abilities from Homologous Models as a Free Lunch. (ICLR 2024)

---

> > ### Comment · Reviewer_6fFE · 2024-11-25
> >
> > Thank you for addressing my concerns thoroughly. I appreciate the thoughtful responses and the approach outlined for merging models trained with different LoRA ranks. The authors have effectively resolved all major points I raised.
> >
> > I have one minor suggestion: While I can empirically understand why merging in V-space might yield better results, the mathematical reasoning behind this choice remains unclear. Providing a stronger theoretical motivation or further insights could enhance the overall understanding and impact of the approach.

---

### Official Review · Reviewer_AJPw · 2024-11-03

**Soundness:** 3
**Presentation:** 3
**Contribution:** 3
**Rating:** 6
**Confidence:** 3

**Summary:**

This paper introduces a new technique to merge the parameters of different LoRAs onto the same model weights. The authors first show that CKA representations better align with the ability to merge model weights, both for fully fine-tuned approaches as well as LoRA-based ones. Then, they propose to use SVD to align the subspaces of different LoRAs. After doing so, the resulting matrices can be merged into a single model by applying previous techniques. The authors show that their approach better merges LoRA weights into a single model, for both vision tasks and language tasks.

**Strengths:**

1. The authors show that CKA representations seem to align with model merging abilities, without some limitations given by orthogonality approaches.
2. The authors propose to merge the LoRA weights after SVD, and showcase better performance than using existing full-rank approaches.
3. The authors propose to evaluate merged models on a multi-task benchmark that they obtained by combining the individual datasets in Ilharco et al. (2023).

**Weaknesses:**

1. While the performance of KnOTS-TIERS is usually significantly better than TIERS, it is not the case for DARE-TIERS. This is not discussed in the paper, and it would be good to understand this behavior.
2. While the CKA alignment given by using KnOTS is significantly better than by the original LoRA weights, the performance improvements (e.g. on DARE-TIERS) are less pronounced, leaving the reader wondering whether CKA is indeed a good-enough metric for weight alignment.
3. The authors should down-weight their “novelty” contributions towards creating a new benchmark by simply putting together the datasets of a previous benchmark. The benchmark itself is a contribution of the paper, but not one that the reviewer feels should be stressed as much. For example, you could present it as a useful extension rather than a major novel contribution.

**Questions:**

1. Can you analyze and discuss potential reasons for the discrepancy in performance gains between KnOTS-TIES and KnOTS-DARE-TIES?
2. Why is the better CKA alignment given by KnOTS not resulting in better performance with KnOTS-DARE-TIERS? Are there additional analyses you could run to better understand the relationship between CKA alignment and performance across different merging method?
3. Can you provide more insights into why row-wise KnOTS does not work?

---

> ### Author Response · Authors · 2024-11-23
> **Author response 1/2 to reviewer AJPw**
>
> We thank the reviewer for their detailed feedback and insightful questions. Please find our responses below.
>
> > **Weakness 1 and Question 1**: While the performance of KnOTS-TIERS is usually significantly better than TIERS, it is not the case for DARE-TIERS. This is not discussed in the paper, and it would be good to understand this behavior.
>
> We thank the reviewer for this good question. We posit that this may be inherently due to the type of pruning used in DARE, compared to TIES. Specifically, DARE randomly prunes elements from the task-updates, while TIES argues that their strategy to prune elements with low-magnitude does not degrade performance by a lot. DARE may elect to prune otherwise significant elements in the task-update which TIES would not select. This may compromise the information preserved in the task-update and yield a poorer overall merged model. This would also inhibit the effectiveness of KnOTS, as critical information in each model would be removed, despite its alignment.
>
> We validate this hypothesis by comparing the performance of a model transformed by our best KnOTS-TIES configuration, and one that is transformed by our best KnOTS-DARE-TIES configuration, before it is linearly combined with others to create a merged model. We conduct this experiment on each of our LoRA Rank 16 ViT-B/32 models from our eight vision task-setting, evaluating each transformed model only on the dataset it was finetuned on. The table below summarizes the results:
> | Transformation  | Avg. Acc.  |
> |-----------------|------------|
> | None            | 84.1       |
> | KnOTS-TIES      | 78.9       |
> | KnOTS-DARE-TIES | 74.5       |
>
> “None” refers to the original finetuned models, and “Avg. Acc.” is the average accuracy of each transformed model evaluated on its respective dataset. Overall, we observe that models transformed by DARE lose a significant amount of average performance (-8.6%) compared to the original models, which may inhibit the merged model’s ultimate capability. In contrast, TIES preserves individual model performances, enabling better merged models.
>
> > **Weakness 2**: While the CKA alignment given by using KnOTS is significantly better than by the original LoRA weights, the performance improvements (e.g. on DARE-TIERS) are less pronounced, leaving the reader wondering whether CKA is indeed a good-enough metric for weight alignment.
>
> We note that CKA is an established metric for measuring the structural alignment between the layers of different models [1, 2]. Similarly, it has been argued in the merging community that models whose layers process activations similarly are easier to merge [3]. We follow these works by measuring the alignments between the layers of the models we wish to merge, and to the best of our knowledge, only use the CKA in the manner it was intended.
>
> > **Question 2**: Why is the better CKA alignment given by KnOTS not resulting in better performance with KnOTS-DARE-TIERS? Are there additional analyses you could run to better understand the relationship between CKA alignment and performance across different merging methods?
>
> Please see our response to "Weakness 1" where we have discussed our analysis between KnOTS-DARE-TIES and KnOTS-TIES. Regarding CKA, we argue that it is primarily valuable in conjunction with individual model performance. If a merging method significantly inhibits the performance capabilities of a base-model, it can be less useful because the quality of the merged model depends on the quality of the models involved in the merge. Thus, we argue the CKA is best utilized to compare approaches which preserve the functional capabilities of their underlying models. An example of this can be found in Fig 2b and Fig 2c. Both figures showcase the degree to which the same two models are aligned in the activation space and V-space respectively.
>
> *References* \
> [1] Kornblith et al., Similarity of Neural Network Representations Revisited. (ICML 2019) \
> [2] Raghu et al., Do Vision Transformers See Like Convolutional Neural Networks? (NeurIPS 2021) \
> [3] Stoica & Bolya et al., ZipIt! Merging Models from Different Tasks without Training (ICLR 2024)

---

> ### Author Response · Authors · 2024-11-23
> **Author response 2/2 to reviewer AJPw**
>
> > **Weakness 3**: The authors should down-weight their “novelty” contributions towards creating a new benchmark by simply putting together the datasets of a previous benchmark. The benchmark itself is a contribution of the paper, but not one that the reviewer feels should be stressed as much. For example, you could present it as a useful extension rather than a major novel contribution.
>
> We acknowledge the reviewer’s thoughts. We agree it is a secondary contribution, and have updated our PDF to make this more clear.
>
> > **Question 3**: Can you provide more insights into why row-wise KnOTS does not work?
>
> We hypothesize that the order in which task specificity occurs matters. In column-wise KnOTS, the task-specific components of each update are intrinsically aligned to transform inputs onto the same basis governed by $U\Sigma$. This representation significantly increases the CKA between models without affecting their individual performance and thus increases the likelihood of a strong merge. However, row-wise KnOTS transforms inputs to different bases. In this case, the CKA between models is equivalent to that of Fig 2b., and we posit that this decreases the likelihood of successful merges.

---

> > ### Comment · Reviewer_AJPw · 2024-11-27
> >
> > I would like to thank the authors for their responses.
> > They have addressed my comments to a good degree, and it would be great (in my opinion) to add the analysis for DARE vs TIES in the next version of the paper.
> > As such, I will keep my positive recommendation towards acceptance of this work at ICLR.

---

### Official Review · Reviewer_c5rA · 2024-11-04

**Soundness:** 2
**Presentation:** 2
**Contribution:** 2
**Rating:** 5
**Confidence:** 3

**Summary:**

The paper "Model Merging with SVD to Tie the Knots" proposes KnOTS, a method to improve merging of LoRA-finetuned models, which traditionally struggle with alignment compared to fully-finetuned models. KnOTS uses SVD to align task-specific updates into a shared space, making it easier to merge LoRA models effectively with existing techniques. The authors also introduce a benchmark to test whether merged models generalize across tasks, showing that KnOTS boosts merging performance by up to 4.3% across various benchmarks in vision and language.

**Strengths:**

* Model merging, specifically for LoRA models, is an interesting and cutting-edge field, with related techniques being widely proposed and explored in recent years.
* KnOTS shows excellent performance across both vision and language tasks, enhancing merged model effectiveness and enabling better generalization on the newly introduced benchmark for multi-task data.

**Weaknesses:**

* As far as I know, merging LoRA models using SVD is not a new technique; implementations have long been available in some open-source libraries and are widely used. Therefore, the innovation in this paper is questionable and appears limited. I'd like to know what are the differences and advantages of the techniques proposed in this paper compared to the SVD-based merging techniques in these libraries?

* The experimental work in this paper is insufficient and does not meet a certain standard; more experimental data is needed to support the authors' announced conclusions. Additionally, the writing quality needs improvement.

**Questions:**

See in Weaknesses.

---

> ### Author Response · Authors · 2024-11-23
> **Author response to reviewer c5rA**
>
> We appreciate and thank the reviewer for their feedback and comments. Please find our responses below.
>
> > **Weakness 1**: As far as I know, merging LoRA models using SVD is not a new technique; implementations have long been available in some open-source libraries and are widely used. Therefore, the innovation in this paper is questionable and appears limited. I'd like to know what are the differences and advantages of the techniques proposed in this paper compared to the SVD-based merging techniques in these libraries?
>
> To the best of our knowledge, KnOTS is a novel method, with significant gains over existing baselines. We kindly request the reviewer to provide references or citations to works that we may have missed.
>
> Specifically, we are unaware of any SVD-based merging techniques. We note that there exists a “ties-svd” method in the popular Hugging Face (HF) library, however, despite the similar name, **KnOTS is completely unrelated**. The source code ([can be found here](https://github.com/huggingface/peft/blob/v0.13.0/src/peft/tuners/lora/model.py#L706)) demonstrates that HF employs the SVD to decompose an already merged LoRA parameter (e.g., with “ties”) back into the LoRA A and B matrices. Thus, the SVD is not involved in the merging process. In contrast, KnOTS explicitly employs the SVD to align and merge models. Given their independence, the HF SVD method can be seamlessly added to KnOTS if desired, yielding a new & unique configuration: “KnOTS-ties-svd.” To ameliorate any confusion, we have added this discussion in the last paragraph of Section 2 in the updated version of our paper.
>
> > **Weakness 2**: The experimental work in this paper is insufficient and does not meet a certain standard; more experimental data is needed to support the authors' announced conclusions. Additionally, the writing quality needs improvement.
>
> We compare KnOTS against baselines on standard vision and language tasks. We employ KnOTS on models of diverse sizes, ranging from ViT-B/32 to ViT-L/14 and the very large Llama3-8B. We introduce a new challenging benchmark for studying the extent to which merged models are general, and study KnOT’s robustness to incrementally merging models. We respectfully contend that our experiments are extensive and as the reviewer notes, KnOTS achieves “excellent performance” across these settings.
>
> Furthermore, we have updated the pdf with a new experiment studying merging models across changing LoRA ranks. Consistent with all our other experiments, KnOTS performs significantly better across ranks. Together, these highlight the robustness of KnOTS across all our settings.
> Regarding writing quality, we would like to note that all other reviewers found the soundness and presentation of our work to be strong, awarding it a high rating of 3. However, if the reviewer has specific suggestions regarding sections that may lack clarity, we would be happy to revise our paper to further improve it.

---

> > ### Comment · Reviewer_c5rA · 2024-11-27
> > **Thanks for your response**
> >
> > Thank you for your response. After carefully reviewing your reply, I realized that I indeed had some misunderstandings earlier. I also noticed the improvements in the overall quality of the article through the revised PDF. As a result, I have increased my score.

---

### Official Review · Reviewer_RYg4 · 2024-11-05

**Soundness:** 3
**Presentation:** 3
**Contribution:** 3
**Rating:** 8
**Confidence:** 3

**Summary:**

The paper titled "Model Merging with SVD to Tie the Knots" explores the challenge of merging Low-Rank Adaptation (LoRA) finetuned models. While model merging has shown success in combining fully-finetuned task-specific models, the same methods often fail when applied to LoRA finetuned models due to misaligned weight structures. To address this, the authors introduce KnOTS, a technique that employs Singular Value Decomposition (SVD) to align LoRA model weights, thereby improving the effectiveness of existing merging methods. KnOTS demonstrates up to a 4.3% improvement in merging performance across vision and language benchmarks. Additionally, the paper presents a new benchmark for assessing the generality of merged models, which evaluates their performance on a joint dataset combining multiple tasks.

**Strengths:**

1.  The concept of using SVD to align model weights for improved merging is novel and practical, addressing a previously unexplored limitation in merging LoRA models.
2. The paper is generally well-structured, with clear descriptions of both the KnOTS methodology and experimental setups.

**Weaknesses:**

1. It would be beneficial if the authors could validate the effectiveness of the method on larger LLMs, such as LLaMA and Qwen2, for more in-depth evaluation.
2. Although the method is effective, the improvements are limited.

**Questions:**

Please check the weakness.

---

> ### Author Response · Authors · 2024-11-23
> **Author Response to Reviewer RYg4**
>
> We thank the reviewer for their feedback and questions! Please find our responses below.
>
> > **Weakness 1**: It would be beneficial if the authors could validate the effectiveness of the method on larger LLMs, such as LLaMA and Qwen2, for more in-depth evaluation.
>
> We thank the reviewer for the suggestion to explore larger LLMs such as LLaMA. We note that we do include Llama3-8B in our experiments (see last paragraph of Section 5.2). Even with the larger models, KnOTS-TIES continues to improve over TIES by nearly 3%. Our experiments in Section 5.2 demonstrate consistent performance improvement across model scales.
>
> > **Weakness 2**: Although the method is effective, the improvements are limited.
>
> We show that KnOTS consistently outperforms prior merging approaches across multiple model scales (Section 5.2), multiple standard benchmark settings (Section 5.1), as well as our new challenging benchmark (Section 5.3) and an incremental merging setting (Section 5.4).
>
> Furthermore, we have updated the pdf with a new experiment studying merging models across changing LoRA ranks. Consistent with all our other experiments, KnOTS performs significantly better across LoRA ranks. KnOTS aligns models without requiring access to data or gradients, making the approach more scalable. We contend that this consistent improvement and our method's ease of use make KnOTS results significant.

---

> > ### Comment · Reviewer_RYg4 · 2024-11-26
> > **Thanks for your reply, I would maintain my rating.**
> >
> > n/a

---

### Author Response · Authors · 2024-12-04
**Thank You to Our Reviewers**

We would like to thank all the reviewers for their valuable feedback, insights, and time. We're particularly encouraged that upon clarification reviewer c5rA increased their rating by 2 points and observed improvement in the overall quality of our revised work. We also thank reviewers RYg4 and AJPw, who were satisfied with our rebuttal, and reviewer 6fFE for acknowledging that our responses addressed their major concerns. We will incorporate the additional analysis conducted during this rebuttal into our camera-ready submission.

---

### Meta-Review · Area_Chair_ogre · 2024-12-22

**Metareview:**

This paper proposes a method for model-merging that is designed for LoRA-finetuned models. Although methods such as TIES and DARE are well known for model-merging, they do not perform well for LoRA-trained models. The authors show that this is because LoRA-parameters are not well aligned between different tasks (as measured by the centred kernel aligned (CKA)), which is different from fully-finetuned models (which is also intuitive, given that LoRA parameters are trained completely from scratch, whilst finetuned models typically all use the same pretraining). To address this problem, the authors first align parameters from different tasks by performing a singular value decomposition (SVD) across all of the tasks. After this, existing model-merging approaches (ie TIES, DARE) can be applied on the aligned weights.

Reviewers appreciated that this is a timely problem, and that the method is simple but effective. The authors addressed the reviewers' comments during the rebuttal, and also revised their paper according to the rebuttal. The most negative review, from Reviewer c5rA who claimed that the SVD has already been used to align weights for model mergining turned out to be a misunderstanding. Therefore, the decision is to accept the paper.

**Additional Comments On Reviewer Discussion:**

Please see above. The authors addressed the reviewers' comments during the rebuttal, and also revised their paper according to the rebuttal. The most negative review, from Reviewer c5rA who claimed that the SVD has already been used to align weights for model mergining turned out to be a misunderstanding.

---

### Decision · Program_Chairs · 2025-01-22

Accept (Poster)